# The Mechanical Properties of Poly (Urea-Formaldehyde) Incorporated with Nano-SiO_2_ by Molecular Dynamics Simulation

**DOI:** 10.3390/polym11091447

**Published:** 2019-09-04

**Authors:** Yanfang Zhang, Youyuan Wang, Yudong Li, Zhanxi Zhang

**Affiliations:** The State Key Laboratory of Power Transmission Equipment & System Security and New Technology, Chongqing University, Chongqing 400044, China

**Keywords:** nano-SiO_2_, poly (urea-formaldehyde), molecular dynamics simulation, mechanical properties, hydrogen bonds

## Abstract

Self-healing materials can promote the sustainable reuse of resources. Poly (urea-formaldehyde) (PUF) microcapsules can be incorporated into dielectric materials for self-healing. However, the mechanical properties of PUF microcapsules need to be improved due to insufficient hardness. In this paper, PUF models incorporated with nano-SiO_2_ of different filler concentrations (0, 2.6, 3.7, 5.3, 6.7, 7.9 wt.%) were designed. The density, the fractional free volume, and the mechanical properties of the PUF-SiO_2_ models were analyzed at an atomic level based on molecular dynamics simulation. The interfacial interaction model of PUF on the SiO_2_ surface was also constructed to further investigate the interaction mechanisms. The results showed that the incorporation of nano-SiO_2_ had a significant effect on the mechanical properties of PUF. Density increased, fractional free volume decreased, and mechanical properties of the PUF materials were gradually enhanced with the increase of nano-SiO_2_ concentration. This trend was also confirmed by experimental tests. By analyzing the internal mechanism of the PUF–SiO_2_ interfacial interaction, it was found that hydrogen bonds play a major role in the interaction between PUF and nano-SiO_2_. Moreover, hydrogen bonds can be formed between the polar atoms of the PUF chain and the hydroxyl groups (–OH) as well as O atoms on the surface of SiO_2_. Hydrogen bonds interactions are involved in adsorption of PUF chains on the SiO_2_ surface, reducing the distance between PUF chains and making the system denser, thus enhancing the mechanical properties of PUF materials.

## 1. Introduction

Dielectric materials are widely used in modern power grids. However, cracking phenomena occasionally happen due to mechanical, thermal, electrical, and electromagnetic damage occurring in the use process. This may lead to partial discharge, electrical treeing, and even the system failing. Therefore, self-healing methods for materials are gradually becoming more studied. The self-healing microcapsule of dicyclopentadiene (DCPD) as the core material and poly (urea-formaldehyde) (PUF) as the wall material is a research hotspot [1,2,3,4,5]. However, PUF microcapsules have insufficient hardness of the shell wall and are easily broken in the lamination process [6], thus the mechanical properties of the microcapsules’ wall materials need to be improved.

A lot of research has shown that incorporation of nanoparticles into polymers can significantly improve mechanical and thermal properties along with the aging resistance of polymers [7,8,9,10]. Ghorbanzadeh Ahangari et al. studied the effect of nanoparticles on the mechanical properties of PUF composite microcapsules. It was found that the elastic modulus and the hardness of PUF microcapsules were significantly increased by incorporation of nano-Al_2_O_3_ [6]. Fereidoon et al. found thermal and water resistances were improved after modifying PUF microcapsule walls by incorporation of single-walled carbon nanotubes or nano-alumina [11]. Jia et al. found nano-MgO could greatly improve tensile strength, Young’s modulus, and elongation at break of biodegradable poly (L-lactic acid) [12]. 

Furthermore, nano-SiO_2_ has been widely used as inorganic nano-filler to improve the mechanical behavior, the chemical stability, and other properties of polymer materials due to its non-toxic, tasteless, non-polluting features [13,14,15,16]. Malaki et al. analyzed the effect of nano-SiO_2_ on the mechanical properties of acrylic polyurethane coatings. It was found that nano-SiO_2_ additives could significantly improve the adhesive strength and notably increase the micro-hardness as well as the erosion resistance of acrylic polyurethane coatings [17]. Fallah et al. found the concrete mechanical properties and durability improved following the introduction of nano-SiO_2_ [18]. Yang et al. found incorporation of nano-SiO_2_ could significantly increase tensile strength and elongation of sodium alginate [19]. Dil et al. found the mechanical properties of poly (lactic acid)/poly (butylene adipate-co-terephthalate) blends were clearly influenced by incorporation of nano-SiO_2_ [20].

As discussed above, incorporation of nanoparticles can significantly enhance various properties of polymers. The mechanical properties of PUF microcapsules play an important role in determining deformability and durability. However, there are few studies on improving PUF mechanical properties by the addition of nanoparticles. Moreover, the studies concentrate on experiments, and there is a lack of internal mechanism research. Generally, traditional experimental tests are hard to use for observation and study of internal mechanisms because the PUF microstructure and the interaction between nanoparticles and PUF are microscopic phenomena. Hence, in-depth studies of the internal interaction mechanism between PUF and nanoparticles have rarely been conducted.

Molecular dynamics (MD) simulation is used to build molecular models at the atomic level to simulate the structure and the behavior of molecules and to simulate various physical and chemical properties of molecular systems by utilizing computers. MD simulation can obtain important microscopic information that cannot be obtained from experimental methods and can explore internal mechanisms to promote the development of theory and experiments. MD simulation has become a promising tool for studying the properties of polymer materials and the internal mechanism between the polymer matrix and the inorganic nanoparticles at the atomic level [21,22,23,24]. Wei et al. investigated the effects of composition ratios on the properties of polymer blend membranes by MD simulation and experiments [25,26]. The effects of nano-SiO_2_ on the mechanical properties of polymer composites were evaluated at an atomic level [27,28]. 

The polymer microstructure and interfaces for adsorption of different species are important for revealing the internal interaction mechanism [29,30,31]. Lai et al. found that the interfacial mechanical behavior between osteopontin peptide and the hydroxyapatite surface was governed mainly by the attractive electrostatic interaction between some acidic amino acids in osteopontin peptide and calcium in hydroxyapatite [32]. Wang et al. successfully elucidated the essence of the interface interaction by calculating the radial distribution function of hydroxyapatite (110)/ α-n-butyl cyanoacrylate based on MD simulation [33]. Wei et al. studied the interface between polymers and nano-SiO_2_ to reveal the interaction mechanism by analyzing the pair correlation functions [34,35]. Kubyshkina et al. demonstrated that the chemistry at the interface between nanoparticles and the polymer matrix influenced charge dynamics in the polymer nanocomposite. The influence of crystal surface termination on electronic properties of interfaces in MgO-polyethylene nanocomposites was also investigated [36]. Pourrahimi et al. suggested interfacial chemistry and area of the polymer/nanoparticle are important factors that impact the dielectric behavior of nanocomposites [37]. 

Inspired by the above studies, nano-SiO_2_ was selected to be incorporated into the PUF wall material of the self-healing microcapsule to improve its mechanical properties. The effects of various nano-SiO_2_ filler concentrations on PUF density, fractional free volume, and the mechanical properties were studied by molecular dynamics simulation. Meanwhile, the effects of nano-SiO_2_ on the mechanical properties of microcapsules were verified by experimental preparation of microcapsules with different filler concentrations. In addition, the interface between PUF and SiO_2_ was studied by analyzing the pair correlation function, and the internal interaction mechanism was revealed. The work improved the mechanical properties of self-healing PUF microcapsules, demonstrated the performance enhancement mechanism, promoted the development of self-healing dielectric materials, and ultimately provides theoretical guidance for the application of nano-SiO_2_/polymer composites.

## 2. Simulation and Experimental Methods 

### 2.1. Materials Models

#### 2.1.1. PUF Model

The appropriate monomer and degree of polymerization are critically important and directly determine the accuracy of simulation results. The production of urea-formaldehyde polymers is typically carried out in two stages. Under appropriate pH and temperature conditions, a series of reactions between formaldehyde molecules and amino groups of urea molecules lead to the formation of pre-polymers; then, urea-formaldehyde polymers are produced under acidic conditions [38]. The methylene linkage of the main products was selected as the monomer to construct the PUF model. Furthermore, a short molecular chain length does not correctly represent real polymers, while a long molecular chain length may consume a lot of time and lead to difficulties in the computer calculations. In this paper, 20 repeat units were selected to establish the model of the PUF molecular chain (Figure 1). All of the simulation models were constructed using Materials Studio software. 

#### 2.1.2. Nano-SiO_2_ Model

The crystal structure of SiO_2_-quartz (space group P3121) was adopted from the Materials Studio Structural Database with the lattice parameters as follows: a = 4.913 Å, b = 4.913 Å, c = 5.4052 Å, α = 90°, β = 90°, γ = 120°. The cell formula is O_6_Si_3_, and the density is 2.649 g/cm^3^. A spherical SiO_2_ nanoparticle of atomic termination on the surface was constructed by adding hydrogen atoms to unsaturated oxygen atoms and hydroxyl groups to unsaturated silicon atoms to avoid an unsaturated boundary effect (Figure 2). The SiO_2_ nanoparticle contained 32 hydroxyl groups on the surface, and the chemical formula is H_32_O_52_Si_18_. The diameter of the spherical SiO_2_ nanoparticle was 10 Å, considering computer efficiency and specific surface area.

#### 2.1.3. PUF-SiO_2_ Models

To study the interaction mechanism between nano-SiO_2_ and PUF as well as prevent any finite size effect of nanocomposites and account for any particle effect at the bulk level, only one SiO_2_ nanoparticle was embedded in the center of the PUF-SiO_2_ model, and the periodic boundary condition was used. PUF-SiO_2_ models with various filler concentrations of nano-SiO_2_ were established by changing the number of PUF chains. The models are represented by the form of mPUF-nSiO_2_-x% (m is the number of PUF chains; n is the nanoparticle number of spherical nano-SiO_2_; and x% is the concentration of SiO_2_ and is defined by the following equation):(1)$$x\%=\frac{m_{SiO_{2}}}{m_{PUF}}\times 100\%$$
where $m_{SiO_{2}}$ is the mass of nano-SiO_2_, and $m_{PUF}$ is the mass of PUF chains. The densities of the original models were all set to 0.5 g/cm^3^ to ensure the molecular chains had sufficient space to relax and avoid overlapping and entanglement. The detailed parameters of the PUF-SiO_2_ models are shown in Table 1.

#### 2.1.4. PUF–SiO_2_ Interfacial Interaction Model

The PUF–SiO_2_ interfacial interaction model was established to further study the interaction mechanism between PUF and SiO_2_ (Figure 3). The crystal structure of SiO_2_-quartz (space group P3121) was adopted, and the most representative surface of SiO_2_ (1 1 0) with the thickness of 19.652 Å was selected to establish the supercell. Similarly, the unsaturated effect of the SiO_2_ (1 1 0) surface was eliminated by adding hydrogen atoms to unsaturated oxygen atoms and hydroxyl groups to unsaturated silicon atoms. The dimensional parameters of the 8 × 8 supercell were a = 43.24 Å, b = 68.08 Å, c = 21.32 Å, and α = β = γ = 90°. Only one PUF chain was contained in the PUF–SiO_2_ interfacial interaction model. A vacuum layer of 30 Å thickness was placed above the PUF molecular chain to ensure the PUF molecular chain only interacted with one side of the SiO_2_ crystal surface. More than one PUF–SiO_2_ interfacial interaction models with different initial PUF 2-D configurations were constructed to obtain the consistent results. Furthermore, the 2-D configuration shown in Figure 3 was selected to construct more than one PUF–SiO_2_ model to obtain the reliable average values.

### 2.2. Molecular Dynamics (MD) Simulation

Before constructing the PUF-SiO_2_ models, the PUF chain and the spherical SiO_2_ nanoparticles were structure-optimized to obtain stable configurations. The six initial PUF-SiO_2_ models were established by constructing amorphous cells, and structure optimization was also performed to eliminate unreasonable interactions and minimize energy for achieving the most stable state. The smart minimization method was utilized with the convergence tolerance of fine quality. The energy convergence was 1 × 10^−4^ kcal/mol, the displacement was 5 × 10^−5^ Å, and the number of iterations was 5000. The COMPASS force field [39] was applied with group-based electrostatic and Van der Waals simulation methods.

Then, MD simulation was performed in three steps. First, the NVT (constant number (N), volume (V), and temperature (T)) ensemble (T = 298 K) for 100 ps of MD simulation was conducted to release any possible tension. Then, the PUF-SiO_2_ models were performed for more than 1500 ps under the NPT (constant number (N), pressure (P), and temperature (T)) ensemble (P = 1 bar, T=298 K) until the density and the energy no longer changed. The equilibration time of NPT-MD simulation depends on the number of atoms in a specific system, and the size and the shape for each system were allowed to vary in order to find the equilibration density. Finally, an additional MD-NVT simulation was conducted for 500 ps, and the trajectory frames were used for results analysis. 

The equilibrium of a system can be judged by the fluctuation of temperature and energy along with simulation time. The whole system has reached equilibrium if temperature and energy fluctuate only between 5% and 10% [40]. The systems in this paper have reached equilibrium judged by temperature and energy fluctuation. For example, time evolution energy and temperature profile for the 37PUF-1SiO_2_-2.6% model during MD simulation are shown in Figure 4. In addition, the errors in the results analysis can be obtained by fluctuation. 

The criteria of all MD simulations are as follows. The COMPASS force field was applied at the fine level of quality. The group based summation method was used for electrostatic and Van der Waals with the cutoff distance of 15.5 Å and a spline width of 1 Å. The temperature for all the simulations was set at 298 K, the time step was 1.0 fs, and the Anderson thermostat and barostat [41] were used to maintain temperature and pressure. The frame outputs of the system were collected every 1000 time steps. The entire calculation process is shown in Figure 5.

Similarly, after the PUF–SiO_2_ interfacial interaction model was established, structural optimization, NVT-MD (600 ps, 298 K), and NVE-MD (600 ps) calculations were performed. 

### 2.3. Preparation Method of DCPD/PUF Microcapsules

The microcapsules consisted of wall materials and core materials. The microcapsules’ wall material was PUF prepared by urea and formaldehyde (reagent mass fraction of 37%). DCPD was used as the core material. Sodium dodecyl benzene sulphonate (SDBS) (99% purity) and resorcinol were respectively used as emulsifier and hardener. Distilled water was used to prepare the aqueous solutions in the whole process. Triethanolamine and HCl solution were used to adjust the pH. The average particle size of the nano-SiO_2_ was 15 nm. All chemicals were used without further purification. Urea, formaldehyde (reagent mass fraction of 37%), DCPD, SDBS, resorcinol, triethylamine, and nano-SiO_2_ were of analytical grade and purchased from Shanghai Aladdin Bio-Chem Technology Co., LTD, Shanghai, China. 

The DCPD/PUF microcapsules were prepared by in situ polymerization in an oil-in-water emulsion. The whole preparation process included two steps. The first step was the preparation of pre-polymer, and the PUF microcapsules could be prepared as the second step. 

At room temperature, urea and 37 wt.% formaldehyde were mixed in a Erlenmeyer flask. The weight ratio of urea and 37 wt.% formaldehyde was 1:2.3. The pH of mixed solution was adjusted to 8–9 with triethanolamine. The system was kept for 1 h with the temperature raised to 73 °C on magnetic stirred equipment. Then, the pre-polymer solution was obtained and cooled to ambient temperature. The solution was colorless and transparent, thus the first step was completed.

Subsequently, 10 g DCPD was dissolved into 100 g deionized water, and 0.5 g emulsifier SDBS was added to form an emulsion. It was allowed to stabilize for 30 min in 40 °C water bath under mechanical agitation, and the pH was adjusted to 7–8 using HCl in the process. Then, 20 g pre-polymer solution and 0.4 g resorcinol were dissolved into the emulsion and kept for 5–10 min. The system was transferred on magnetic stirred equipment. The pH of the emulsion was adjusted slowly to 3.0–4.0 using HCl, and it was covered and slowly heated to the target temperature of 63 °C. After 3 h, the reaction was completed. The obtained microcapsules were filtered and rinsed with deionized water. Then the microcapsules were dried in the drying oven at 25 °C for 24 h. The DCPD/PUF microcapsules without incorporation of nano-SiO_2_ were prepared. 

DCPD/PUF microcapsules incorporated with nano-SiO_2_ (2.6, 3.7, 5.3, 6.7, 7.9 wt.%) were prepared by using the same procedures with the addition of nano-SiO_2_ (0.079, 0.112, 0.161, 0.204, 0.240 g) when adding the pre-polymer. Twenty grams of pre-polymer solution were prepared by 2.5 g of urea. On the basis of Equation (2) and assuming the polymerization reaction as complete, the mass of PUF contained in the pre-polymer solution could be estimated followed by the nano-SiO_2_ content of microparticles. Moreover, the successfully prepared DCPD/PUF microcapsule is shown in Figure 6.

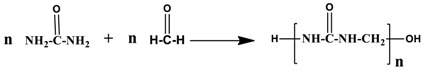
(2)

## 3. Results

### 3.1. Density

After the MD simulation, the final equilibrium models of PUF incorporated with nano-SiO_2_ under various filler concentrations at 298 K were obtained (Figure 7). The densities of the six PUF-SiO_2_ models after equilibration (T = 298 K, P = 1 bar) are illustrated in Table 1. Comparing the densities (1.184 ± 0.002 and 1.307 ± 0.003 g/cm³) for the models of 37PUF-0SiO_2_-0% and 37PUF-1SiO_2_-2.6%, it can be seen that incorporation of nano-SiO_2_ was beneficial to increasing the density of PUF materials. For the models of 37PUF-1SiO_2_-2.6%, 25PUF-1SiO_2_-3.7%, 18PUF-1SiO_2_-5.3%, 14PUF-1SiO_2_-6.7%, and 12PUF-1SiO_2_-7.9%, the densities gradually increased with the increase of filler concentration. In other words, the incorporation of nano-SiO_2_ made the system denser. The reason for this was that interaction force could be formed between PUF chains and nano-SiO_2_. The interaction force made the PUF chain absorb on the surface of the nano-SiO_2_ and reduced the distance between the PUF chains.

### 3.2. Fractional Free Volume

The void space within molecules was considered as the free volume. The free volume of the PUF model incorporated with nano-SiO_2_ was studied using a hard spherical probe. The Connolly surface is calculated when the probe molecule rolls over the Van der Waals surface, and the free volume is determined as the volume on the side of the Connolly surface without atoms. However, free volumes of different models cannot be directly compared. The fractional free volume (FFV) has more computational significance. The microscopic morphological structures of different models were evaluated by calculating their FFVs with a probe radius of 1.0 Å. The FFV is defined by [42]:(3)$$FFV=\frac{V_{f}}{V_{f}+V_{o}}\times 100\%$$
where $V_{f}$ is the free volume and $V_{o}$ is the volume occupied by the polymer materials. 

The results of free volumes, occupied volumes, and FFVs for different PUF-SiO_2_ models under various filler concentrations are shown in Table 2. Comparing the FFVs (28.01 ± 0.07 and 20.31 ± 0.03%) for the models of 37PUF-0SiO_2_-0% and 37PUF-1SiO_2_-2.6%, it can be seen that incorporation of nano-SiO_2_ was beneficial to reducing the FFV of PUF materials. Moreover, when the filler concentrations of nano-SiO_2_ in PUF-SiO_2_ models were 2.6, 3.7, 5.3, 6.7, and 7.9 wt.%, the FFVs were 20.31 ± 0.03, 16.66 ± 0.08, 13.70 ± 0.12, 12.38 ± 0.09, and 9.56 ± 0.07%, respectively. This indicated FFVs of the six PUF models incorporated with nano-SiO_2_ gradually decreased with the increase of filler concentration. The trend of FFVs is also shown in Figure 8. It is consistent with the density results of different PUF-SiO_2_ models in Table 1. The main reason behind this was that a strong interaction force could be formed between PUF chains and nano-SiO_2_. The interaction force reduced the distance between molecules and made the system denser. Therefore, the incorporation of nano-SiO_2_ was beneficial to the increase of densities and the decrease of FFVs.

### 3.3. Mechanical Properties

#### 3.3.1. Micromechanical Properties

The effect of incorporating nano-SiO_2_ on the mechanical properties of PUF was analyzed by calculating the micro mechanical properties of the six PUF-SiO_2_ models. The micro mechanical properties were obtained by analyzing the atom trajectories of PUF-SiO_2_ models and could not fully represent the macro mechanical properties. However, the feasibility of the micro mechanical properties could be verified by comparing the micro mechanical properties with the macro mechanical properties. 

In the simulation, the mechanical properties of the PUF nanocomposites were calculated by adopting the constant-strain minimization method. The static method was carried out after structure optimization and MD simulation. The application of the strain was performed by uniformly expanding the simulation domain in the direction of the deformation and re-scaling the coordinates of the molecules to fit within the new dimensions. After each increment of the applied strain, the potential energy of the structure was re-minimized and maintained the lattice parameters fixed [43]. The mechanical properties included Young’s modulus (E), bulk modulus (K), shear modulus (G), Poisson's ratio (γ), and Cauchy pressure ($C_{12}-C_{44}$), among others. Young’s modulus is the ratio of stress to strain and can characterize the stiffness of the material. Young’s modulus is positively related to the stiffness of a material. The larger the Young’s modulus is, the stronger the material’s ability to resist deformation will be. The bulk modulus can characterize the incompressibility of a material. The shear modulus is the ratio of shear stress to shear strain and also reflects the material’s ability to resist deformation. The Cauchy pressure ($C_{12}-C_{44}$) is used to measure the ductility of a material. The greater the value is, the better the ductility of the material will be. Young’s modulus (E), bulk modulus (K), shear modulus (G), and Poisson's ratio (γ) are calculated as follows: (4)$$E=\mu \frac{3\lambda +2\mu }{\lambda +\mu }$$
(5)$$K=\lambda +\frac{2}{3}\mu$$
(6)G=μ
(7)$$\gamma =\frac{1}{2}\frac{\lambda }{\lambda +\mu }$$
where λ and μ are Lamé constants and can be calculated from the elastic coefficients according to the elasticity statistical mechanics [44].
(8)$$\lambda =\frac{1}{3}\left(C_{11}+C_{22}+C_{33}\right)-\frac{2}{3}\left(C_{44}+C_{55}+C_{66}\right)$$
(9)$$\mu =\frac{1}{3}\left(C_{44}^{}+C_{55}+C_{66}\right)$$

The elastic coefficient $C_{ij}$ is calculated as follows:(10)$$C_{ij}=\frac{1}{V}\frac{\partial^{2}U}{\partial \varepsilon_{i}\varepsilon_{j}}=\partial \sigma_{i}/\partial \varepsilon_{j}$$
where V is the domain volume, U is the potential energy of the structure, ε is the strain, and σ is the stress component.

The values of $C_{11}$, $C_{22}$, $C_{33}$, $C_{44}$, $C_{55}$, and $C_{66}$ could be extracted from the result file. Therefore, the values of Lamé constants λ and μ could be obtained according to Equations (8) and (9). Moreover, the values of E, K, G, and γ about mechanical properties could be obtained according to Equations (4)–(7). The errors in mechanical properties calculations were computed as mean-square deviations from the average value obtained by averaging all samples and three directions (x, y, z) [45].

The results of the micro mechanical properties are shown in Table 3. It can be seen that Young’s modulus E, bulk modulus K, and shear modulus G increased with the incorporation of nano-SiO_2_ when comparing the models of 37PUF-0SiO_2_-0% with 37PUF-1SiO_2_-2.6%. This indicated that the incorporation of nano-SiO_2_ resulted in the enhancement of mechanical properties of PUF materials. For the data of 37PUF-1SiO_2_-2.6%, 25PUF-1SiO_2_-3.7%, 18PUF-1SiO_2_-5.3%, 14PUF-1SiO_2_-6.7%, and 12PUF-1SiO_2_-7.9%, we know that E, K, and G of PUF materials gradually increased with the increase of nano-SiO_2_ filler concentration, which corresponds to the results of density in Table 1 and FFV in Table 2. The mechanical properties of PUF materials were enhanced when the density increased and the FFV decreased. In addition, the values of Cauchy pressure $C_{12}-C_{44}$ also gradually increased, indicating that the ductility of the PUF material tended to change for the better with the increase of nano-SiO_2_ filler concentration. The Poisson’s ratio was 0.25–0.34, which is in the range of the Poisson’s ratio of plastics (0.2–0.4). To verify the reliability of the computed results, PUF microcapsules containing DCPD were prepared, and the Young’s modulus was tested and described (Section 3.3.2).

From the data in Table 1, Table 2 and Table 3, we know that the incorporation of nano-SiO_2_ was beneficial to the increase of the PUF density, the decrease of the PUF FFV, and the enhancement of the PUF mechanical properties. Moreover, densities gradually increased, FFVs gradually decreased, and mechanical properties gradually enhanced with the increase of nano-SiO_2_ concentration. The reason for this was the interaction force between the PUF chains and th enano-SiO_2_. There was an increase in density, a decrease in FFV, and an enhancement in the mechanical properties of PUF materials due to the interaction force. 

#### 3.3.2. Mechanical Properties of DCPD/PUF Microcapsules

The self-healing microcapsules consisted of the DCPD core material and PUF wall material. DCPD/PUF microcapsules incorporated with nano-SiO_2_ under filler concentrations of 0, 2.6, 3.7, 5.3, 6.7, and 7.9 wt.% were prepared. Nanoindentation tests were carried out on the DCPD/PUF microcapsules, and the values of the corresponding Young’s modulus were obtained. The nanoindentation tests were performed on a nanomechanical test instrument (Nano Test Vantage, Micro Materials, Wrexham, UK). Microcapsules were fixed on a glass slide. The tip approached the microcapsule at a certain rate. After engaging the tip with the microcapsule surface, the load was increased at the loading rate until the predefined maximum load was achieved. The maximum peak loads of 0.07 mN and the loading rate of 0.007 mN/s were applied for indentation experiments. Next, to minimize the time-dependent plastic effect, the maximum load remained constant for 10 s. Then, in the unloading segment, the tip was withdrawn from the microcapsule surface at the same rate. To obtain reliable results, Young’s modulus of microcapsules was the average value of 20 microcapsules for each filler concentration.

The experimental values of Young’s modulus were 1.70 ± 0.46, 3.20 ± 0.62, 4.71 ± 0.53, 5.18 ± 0.58, 5.62 ± 0.64, and 6.43 ± 0.47 GPa for the PUF microcapsules with nano-SiO_2_ filler concentrations of 0, 2.6, 3.7, 5.3, 6.7, and 7.9 wt.%, respectively. The calculated values of Young’s modulus were 6.28 ± 0.38, 7.56 ± 0.21, 7.92 ± 0.47, 8.09 ± 0.33, 8.94 ± 0.62, and 10.75 ± 0.67 GPa. The Young’s modulus of the calculated values and the experimental values had the same trend (Figure 9). Both the experimental values and the calculated values of Young’s modulus gradually increased with the increase of nano-SiO_2_ concentration. This also verified the feasibility of the simulation calculation.

From the analysis of density, FFV, and mechanical properties of PUF materials above, we know that the incorporation of nano-SiO_2_ was beneficial to the increase of the PUF density, the decrease of the PUF FFV, and the enhancement of the PUF mechanical properties. Moreover, densities gradually increased, FFVs gradually decreased, and mechanical properties gradually enhanced with the increase of nano-SiO_2_ concentration. The reason was speculated to be the interaction force between PUF chain and nano-SiO_2_. Thus, the internal mechanism of the interaction between PUF and nano-SiO_2_ needed to be revealed.

## 4. Mechanism Analysis

### 4.1. Interfacial Binding Energy

The binding energy and the hydrogen bond energy of a PUF chain on the SiO_2_ crystal surface can reflect the interaction. The binding energy can be calculated by:(11)$$E_{binding}=-E_{\mathrm{int}eraction}=-\left(E_{PUF/SiO_{2}}-E_{PUF}-E_{SiO_{2}}\right)$$
(12)$$E_{hydrogen\;bond}=E_{hydrogen\;bond}{}_{\left(PUF/SiO_{2}\right)}-E_{hydrogen\;bond\left(PUF\right)}-E_{hydrogen\;bond\left(SiO_{2}\right)}$$
where $E_{binding}$, $E_{\mathrm{int}eraction}$, and $E_{hydrogen\;bond}$ are, respectively, binding energy, interaction energy, and hydrogen energy between PUF and the SiO_2_ crystal surface, $E_{PUF/SiO_{2}}$ and $E_{hydrogen\;bond\left(PUF/SiO_{2}\right)}$ are, respectively, total energy and hydrogen bond energy of the PUF–SiO_2_ system, $E_{PUF}$ and $E_{hydrogen\;bond\left(PUF\right)}$ are, respectively, total energy and hydrogen bond energy of the PUF chain without the SiO_2_ crystal surface, and $E_{SiO_{2}}$ and $E_{hydrogen\;bond\left(SiO_{2}\right)}$ are, respectively, total energy and hydrogen bond energy of the SiO_2_ crystal surface without the PUF chain. All the energies of each part (PUF–SiO_2_, separated PUF chain, separated SiO_2_ crystal surface) were calculated after structural optimization and MD simulation. Moreover, the Dreiding force field was used in the energy calculation process, and by the force field, the hydrogen bond energies of each part could be obtained. The average interaction energy and hydrogen bond energy between PUF chain and SiO_2_ are shown in Table 4. The interaction energy of PUF–SiO_2_ was −66.8 ± 0.2 kcal/mol based on Equation (11), thus the binding energy was 66.8 ± 0.2 kcal/mol. Moreover, the hydrogen bond energy was −45.1 ± 0.8 kcal/mol based on Equation (12). From the data in Table 4, we know that the hydrogen bonds interaction played a major role in the interaction between the PUF chain and the SiO_2_ crystal surface.

### 4.2. The Interfacial Hydrogen Bond Number

A hydrogen atom is covalently bonded to an atom X (F, O, N, etc.) with large electronegativity and a small radius. If the hydrogen atom comes close to the electron-negative atom Y (which can also be the same as X), a special intermolecular interaction in the form of XH...Y will be formed with the hydrogen as a medium between X and Y, called a hydrogen bond. Hydrogen bond is defined by the geometry rule of the relative position between two molecules (the hydrogen-acceptor distance $r_{HA}$ ≤ 3.5 Å and the donor-hydrogen-acceptor angle β ≥ 90°), as shown in Figure 10. The oxygen atom of the hydroxy group is called the hydrogen bond donor, because it is "donating" its hydrogen to the nitrogen. The nitrogen atom is called the hydrogen bond acceptor, because it is "accepting" the hydrogen from the oxygen. 

The interfacial hydrogen bond number between the PUF chains and the nano-SiO_2_ surface of PUF-SiO_2_ models is calculated by the following expression [46]:(13)$$N_{\mathrm{int}erface}=N_{total}-N_{PUF}-N_{nano-SiO_{2}}$$
where $N_{\mathrm{int}erface}$ is the interfacial hydrogen bond number between the PUF chains and the nano-SiO_2_ surface of the PUF-SiO_2_ model, $N_{total}$ is the total hydrogen bond number of the PUF-SiO_2_ model, $N_{PUF}$ is the number of PUF chain intramolecular and intermolecular hydrogen bonds, and $N_{nano-SiO_{2}}$ is the hydrogen bond number in nano-SiO_2_. It can be seen from Table 5 that $N_{total}$ for the model of 37PUF-1SiO_2_-2.6% was more than that for the model of 37PUF-0SiO_2_-0%. The reason for this was that there were hydrogen bonds between the PUF chains and the nano-SiO_2_ due to the presence of nano-SiO_2_. For the models of 37PUF-1SiO_2_-2.6%, 25PUF-1SiO_2_-3.7%, 18PUF-1SiO_2_-5.3%, 14PUF-1SiO_2_-6.7%, and 12PUF-1SiO_2_-7.9%, $N_{total}$ gradually decreased with the increase of nano-SiO_2_ concentration due to the decrease in the number of PUF chains. In addition, $N_{PUF}$ also gradually decreased due for the same reason. Moreover, the decrease of PUF chains had a significant effect on $N_{\mathrm{int}erface}$. However, it was meaningless to directly compare the $N_{\mathrm{int}erface}$ of different PUF-SiO_2_ models. There were different numbers of PUF chains surrounding the SiO_2_ nanoparticle in the six PUF-SiO_2_ models. The ratio of $N_{\mathrm{int}erface}/N_{PUF}$ was more valuable for reflecting the interface interactions between the PUF chains and the nano-SiO_2_. The ratio of $N_{\mathrm{int}erface}/N_{PUF}$ gradually increased with the increase of nano-SiO_2_ concentration. This is a good example of the trend in density, FFV, and mechanical properties for the PUF-SiO_2_ models, implying that the interaction force between the PUF and the nano-SiO_2_ gradually increased, and more PUF chains were adsorbed on the surface of the nano-SiO_2_, reducing the distance between PUF chains, increasing the densities, and enhancing the mechanical properties of the PUF materials.

### 4.3. Interfacial Interaction Mechanism

Hydrogen bond interaction plays a major role in PUF–SiO_2_ interaction energy. Hydrogen bonds exist between the polar atoms in a PUF chain and the atoms on the surface of nano-SiO_2_. Therefore, to further explore the interaction mechanism of PUF–SiO_2_, the pair correlation function (PCF) between the PUF chain and the nano-SiO_2_ was studied by analyzing the equilibrium trajectory file. PCF is the appearance possibility of other particles at a distance r from a given particle and is used to characterize the interaction between non-bonded atoms. The possibility is represented by g(r).
(14)$$g_{A-B}\left(r\right)=\left(\frac{n_{B}}{4\pi r_{2}dr}\right)/\left(\frac{N_{B}}{V}\right)$$
where $n_{B}$ is the number of B atoms that surround A atoms at distance r, $N_{B}$ is the total number of B atoms, and V is the volume of the entire system.

The schematic diagram of hydrogen bonds between the PUF chain and the nano-SiO_2_ is shown in Figure 11. There were hydroxyl groups (–OH), O atoms, and Si atoms on the surface of nano-SiO_2_. There were skeleton C atoms, N atoms, O atoms of carbonyl groups (C=O), and H atoms in the PUF chain. According to the definition of a hydrogen bond, it was speculated that there were hydrogen bonds between hydroxyl groups (–OH) on the surface of nano-SiO_2_ and N atoms as well as O atoms in the PUF chain. In addition, hydrogen bonds existed between O atoms on the nano-SiO_2_ surface and –HN in the PUF chain.

Therefore, hydrogen bonds between the PUF chain and the nano-SiO_2_ were studied from PCFs by analyzing the equilibrium trajectory files, as in Figure 11. In these PCFs, OH(SiO_2_), O(SiO_2_), and Si(SiO_2_) respectively represent hydroxyl groups (–OH), O, and Si atoms on the surface of nano-SiO_2_. C(PUF), N(PUF), O(PUF), and H(PUF) respectively represent skeleton C, N, O, and H atoms in the PUF chain.

In general, intermolecular interactions include chemical bonding, hydrogen bonding, and Van der Waals forces. Chemical bonds and hydrogen bonds are found when the position of the peak in PCF is less than 3.5 Å. This indicates Van der Waals force when the position of the peak in PCF is more than 3.5 Å.

PCFs of the PUF–SiO_2_ interfacial interaction are shown in Figure 12. According to Figure 12a, there were three main peaks of g(r) appearing at r = 1.75, 2.75, and 4.25 Å for OH(SiO_2_)-O(PUF). It was predicted that hydroxyl groups (–OH) on the surface of nano-SiO_2_ interacted with O atoms in the PUF chain mainly by hydrogen bonds interactions at r = 1.75 and 2.75 Å. The peak at r = 4.25 Å was in the effective range of Van der Waals forces. In addition, the peak values at r = 1.75 and 2.75 Å were bigger than the value at r = 4.25 Å, which indicated hydrogen bonds interactions accounted for a bigger portion and Van der Waals forces accounted for a smaller portion of OH(SiO_2_) interactions with O(PUF).

The two main g(r) peaks for OH(SiO_2_)-H(PUF) appeared at r = 3.25 and 4.25 Å. The peak at r = 3.25 Å represented hydrogen bonds, while the peak at r = 4.25 Å represented Van der Waals forces. In fact, the hydrogen bonds of OH(SiO_2_)-H(PUF) included OH(SiO_2_)-NH(PUF), O(SiO_2_)-HN(PUF), and so on. 

The two main g(r) peaks for OH(SiO_2_)-N(PUF) appeared at r = 1.75 and 4.25 Å. The value at r = 1.75 Å was smaller, which indicated only a small portion of N(PUF) atoms could interact with the hydroxyl groups (–OH) on the SiO_2_ surface by hydrogen bonding, and most of them formed Van der Walls force.

The first g(r) peak for OH(SiO_2_)-C(PUF) appeared at r = 3.75 Å. It indicated hydroxyl groups (–OH) on the SiO_2_ surface interacting with C atoms in the PUF chain, all by Van der Waals forces.

For Figure 12b, the two main g(r) peaks for O(SiO_2_)-H(PUF) appeared at r = 2.75 and 9.75 Å. The value at r = 2.75 Å was very small, indicating a very small portion of the H(PUF) atoms could interact with the O atoms on the SiO_2_ surface by hydrogen bonding. Actually, hydrogen bonds occurred between O(SiO_2_) and HN(PUF). The other peaks of O(SiO_2_)-H(PUF) and all peaks for O(SiO_2_)-C(PUF), O(SiO_2_)-N(PUF), and O(SiO_2_)-O(PUF) were in the range of Van der Waals forces. Moreover, the peak values representing hydrogen bonds were small, thus the O atoms on the surface of SiO_2_ interacted with the PUF chain mainly by Van der Waals forces.

According to Figure 12c, showing PCFs of the Si(SiO_2_)-PUF, all g(r) peaks were very small and appeared at r > 3.5 Å, which indicated that there were no hydrogen bonds but only Van der Waals forces between Si atoms on the surface of the SiO_2_ and the PUF chain.

Lastly, it was found that hydrogen bonds between OH(SiO_2_) and the PUF chain were the main part of hydrogen bonds formation for the PUF–SiO_2_ interfacial interaction according to the height of the data lines for Figure 12a–c. Furthermore, the larger the g(r) peak values were, the stronger the hydrogen bond strength was, thus the hydrogen bond strength followed the order of OH(SiO_2_)...O(PUF) > OH(SiO_2_)...H(PUF) > OH(SiO_2_)...N(PUF) > O(SiO_2_)...HN(PUF).

Briefly, the PCFs of the PUF–SiO_2_ interfacial interaction revealed the interaction mechanism between the PUF chain and the nano-SiO_2_. The hydrogen bonds of the PUF–SiO_2_ interfacial interaction were formed, made PUF chains adsorb on the nano-SiO_2_ surface, reduced the distances between the PUF chains, increased the density, reduced the FFV, and ultimately enhanced the mechanical properties.

## 5. Conclusions

In this paper, the effect of incorporation of nano-SiO_2_ into poly (urea-formaldehyde) (PUF) on the mechanical properties was studied by establishing six PUF models incorporated with nano-SiO_2_ under various filler concentrations. In addition, the interaction mechanism was revealed by constructing one PUF–SiO_2_ interfacial interaction model. The densities, the fractional free volumes (FFVs), and the mechanical properties of the PUF-SiO_2_ models were analyzed. For comparison purposes, DCPD/PUF microcapsules were also prepared.

It was found that there was an increase in density, a decrease in FFV, and an enhancement of the mechanical properties of PUF materials with the incorporation of nano-SiO_2_. Moreover, the density gradually increased, the FFV gradually decreased, and the mechanical properties were gradually enhanced with the increase of nano-SiO_2_ filler concentration. In addition, the Young’s modulus of PUF microcapsules prepared by experiments gradually increased with the increase of nano-SiO_2_ filler concentration. The trends of molecular dynamics simulation and the experimental results were the same. The reason for this was that interaction forces could be formed between PUF chains and nano-SiO_2_. 

The analysis of the energies for the PUF–SiO_2_ interfacial interaction showed that hydrogen bonds played a major role in the interaction between PUF and nano-SiO_2._ The proportion of interfacial hydrogen bonds number became higher and higher with the increase of the nano-SiO_2_ filler concentration. Hydrogen bonds could be formed between hydroxyl groups (–OH) as well as O atoms on the SiO_2_ surface and the polar atoms in the PUF chain according to the analysis of pair correlation function. Hydrogen bonds made PUF chains adsorb on the surface of nano-SiO_2_ and reduced the distances between molecules, increasing the compactness and thereby enhancing the mechanical properties of the PUF materials. It revealed the internal mechanism of nano-SiO_2_ enhancing PUF mechanical properties by molecular dynamics simulation.

This work not only contributes to the understanding of the microstructure and the interaction mechanism of PUF-SiO_2_ nanocomposites, facilitating their in-depth research and design, but also improves the defect of insufficient hardness for DCPD/PUF self-healing microcapsules and promotes the further development of self-healing dielectric materials.

## Figures and Tables

**Figure 1 polymers-11-01447-f001:**
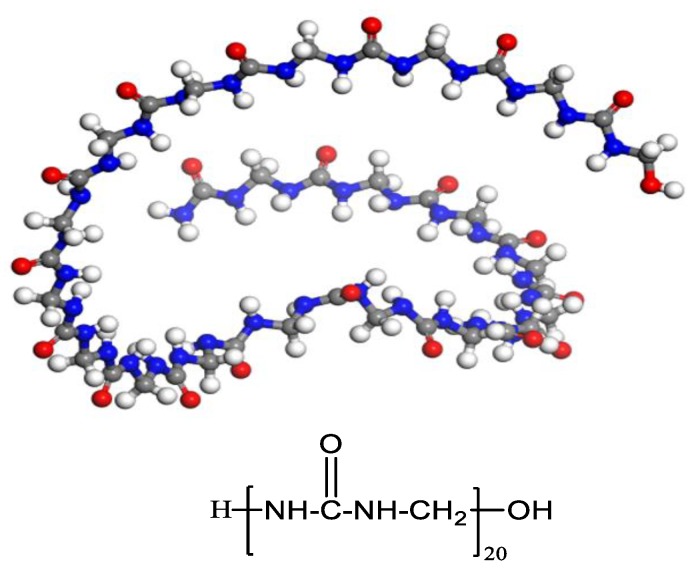
The model of the poly (urea-formaldehyde) (PUF) molecular chain.

**Figure 2 polymers-11-01447-f002:**
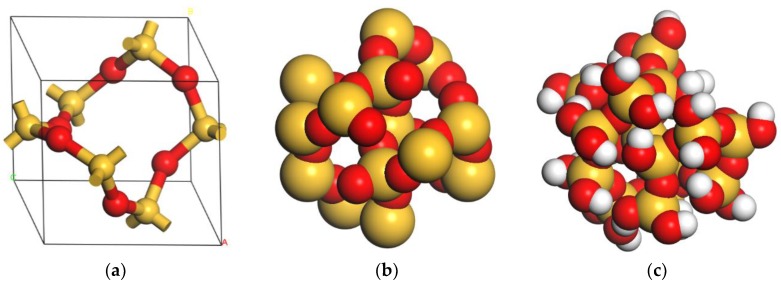
The SiO_2_ models: (**a**) crystal structure of SiO_2_-quartz; (**b**) spherical SiO_2_ nanoparticle with unsaturated bonds on the surface; (**c**) spherical SiO_2_ nanoparticle of atomic termination on the surface.

**Figure 3 polymers-11-01447-f003:**
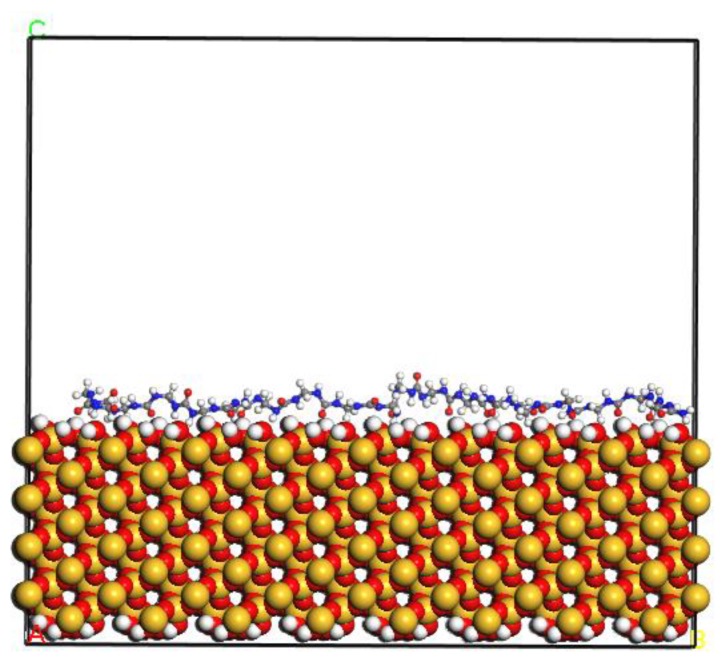
PUF–SiO_2_ interfacial interaction model.

**Figure 4 polymers-11-01447-f004:**
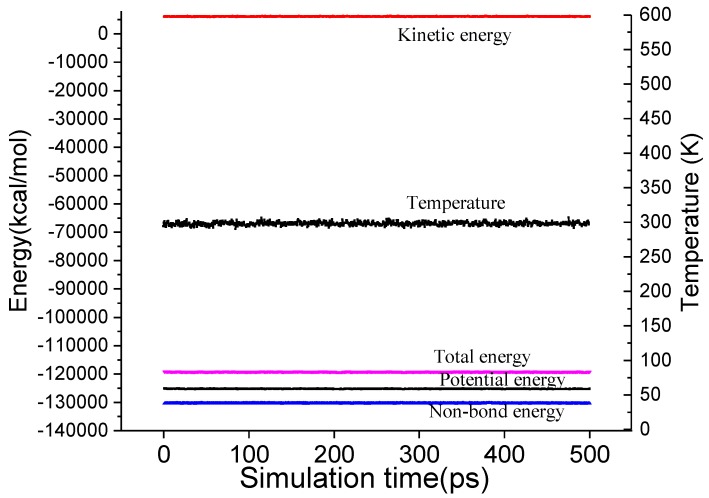
Energy and temperature of 37PUF-1SiO_2_-2.6% model as a function of NVT-molecular dynamics (MD) simulation time.

**Figure 5 polymers-11-01447-f005:**
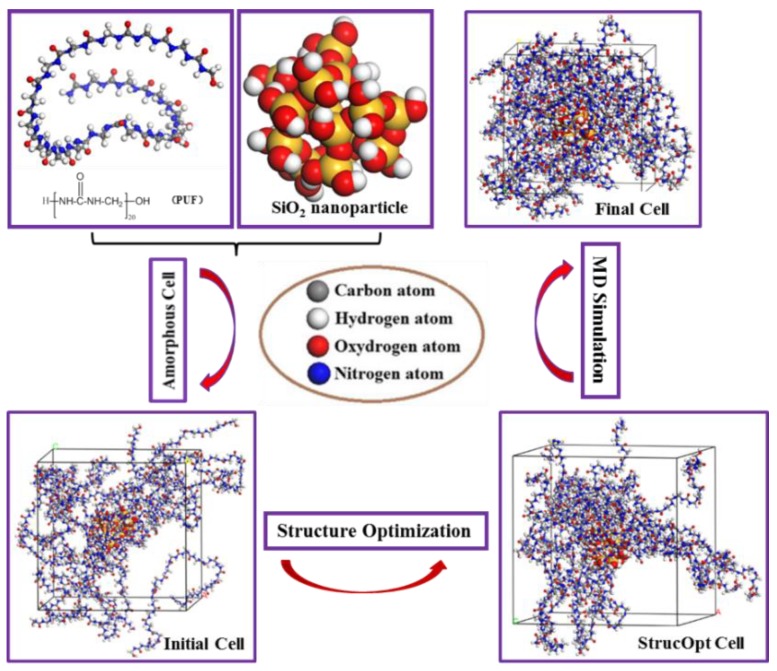
The entire calculation process of the PUF-SiO_2_ model. Here, the MD simulation contains three steps: NVT-MD simulation (T = 298 K, 100 ps); NPT-MD simulation (P = 1 bar, T = 298 K, >1500 ps); NVT-MD simulation (T = 298 K, 500 ps).

**Figure 6 polymers-11-01447-f006:**
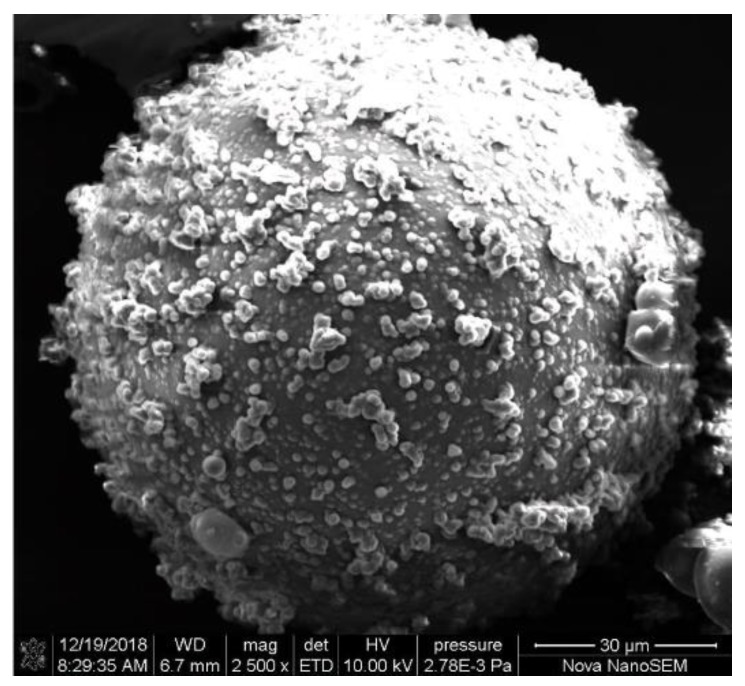
Scanning electron micrograph of dicyclopentadiene (DCPD)/PUF microcapsule.

**Figure 7 polymers-11-01447-f007:**
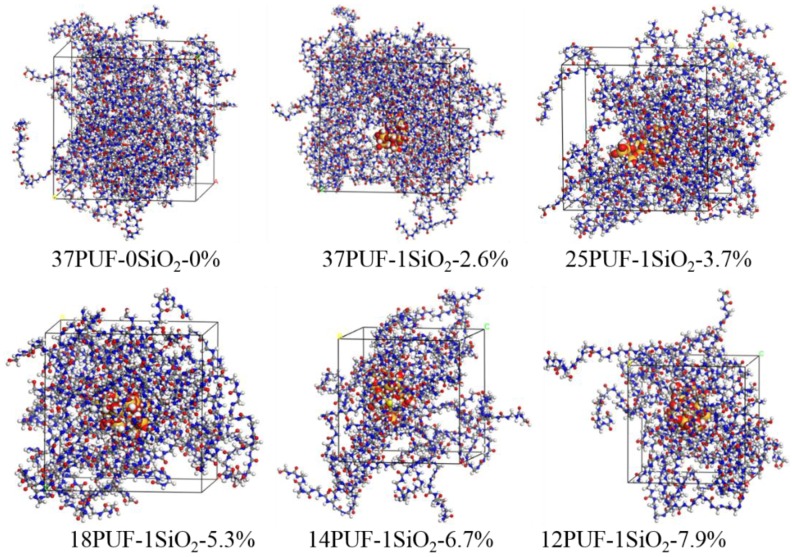
The final equilibrium PUF-SiO_2_ models.

**Figure 8 polymers-11-01447-f008:**
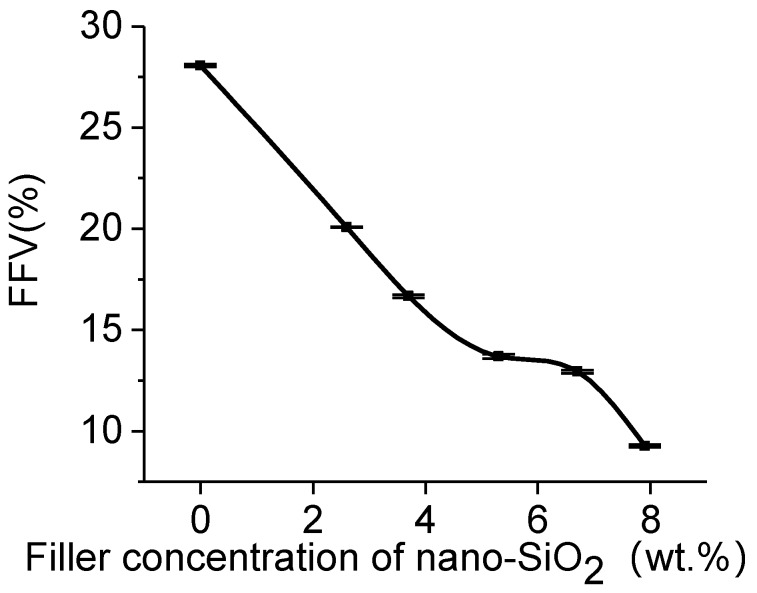
Fractional free volume (FFV) of PUF-SiO_2_ models.

**Figure 9 polymers-11-01447-f009:**
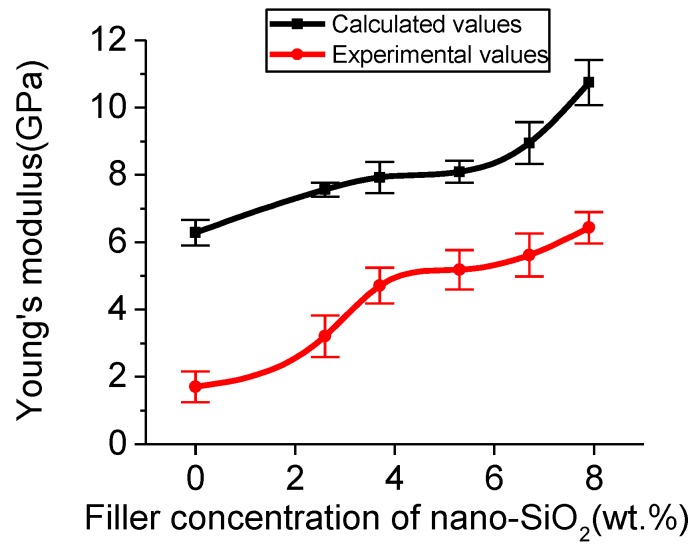
Calculated and experimental values of Young’s modulus for the PUF-SiO_2_ models.

**Figure 10 polymers-11-01447-f010:**
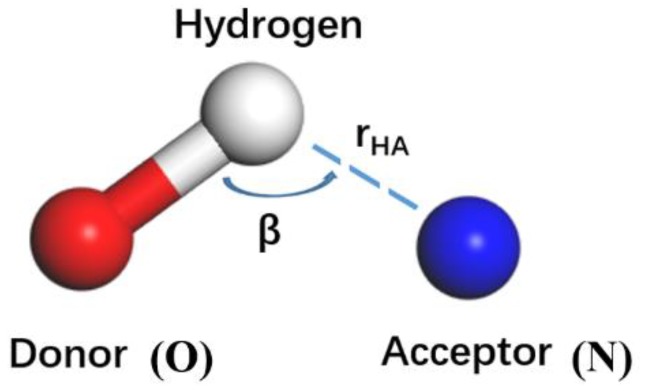
The schematic diagram of a hydrogen bond.

**Figure 11 polymers-11-01447-f011:**
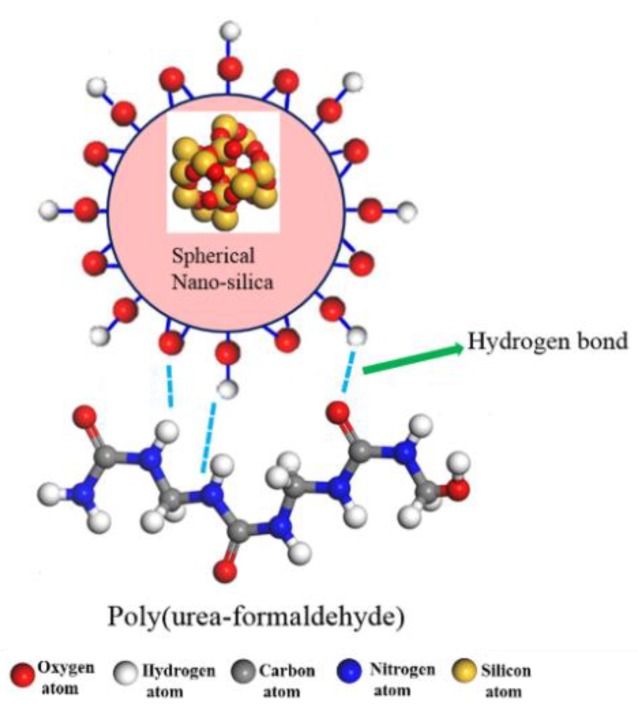
The schematic diagram of hydrogen bonds for the PUF–SiO_2_ interfacial interaction. The PUF chain was constructed by three repeat units.

**Figure 12 polymers-11-01447-f012:**
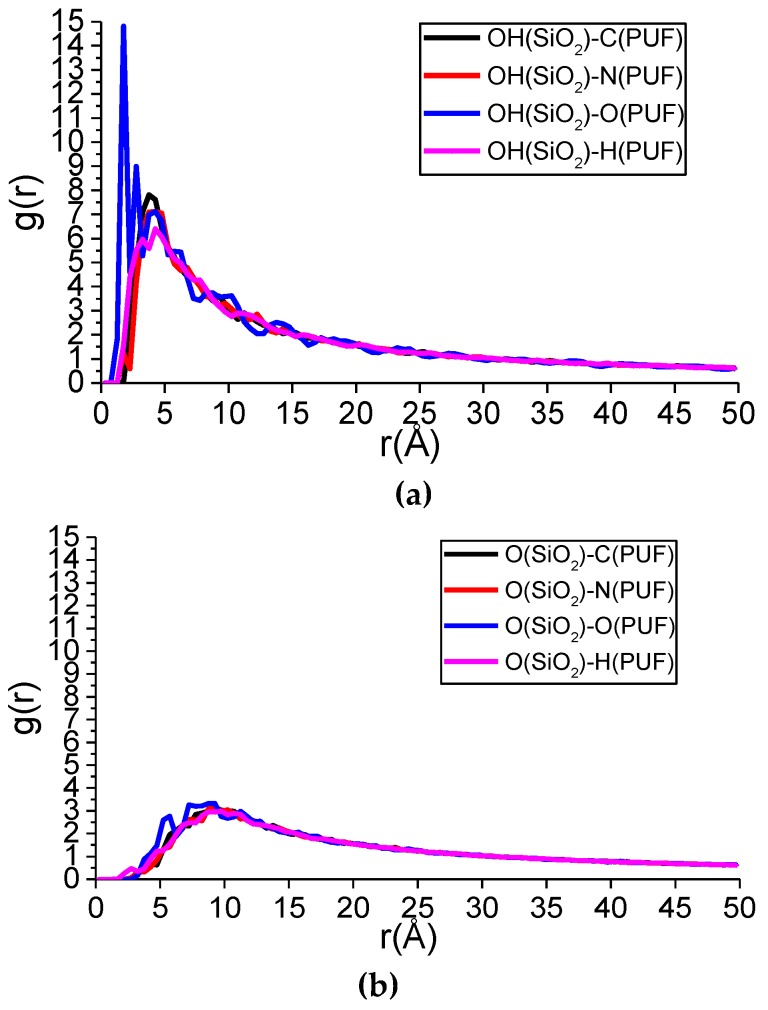
Pair correlation function (PCF) of the PUF–SiO_2_ interface interaction. (**a**) PCFs of OH(SiO_2_)-PUF; (**b**) PCFs of O(SiO_2_)-PUF; (**c**) PCFs of Si(SiO_2_)-PUF.

**Table 1 polymers-11-01447-t001:** The cell parameters of PUF-SiO_2_ models (mean ± standard error).

Model	Before NPT-MD		After NPT-MD	
	Cell Length (Å)	Density (g/cm³)	Cell Length (Å)	Density (g/cm³)
37PUF-0SiO_2_-0%	56.4	0.5	42.30 ± 0.02	1.184 ± 0.002
37PUF-1SiO_2_-2.6%	56.9	0.5	41.28 ± 0.03	1.307 ± 0.003
25PUF-1SiO_2_-3.7%	50.1	0.5	35.81 ± 0.04	1.369 ± 0.004
18PUF-1SiO_2_-5.3%	45.1	0.5	31.87 ± 0.03	1.418 ± 0.003
14PUF-1SiO_2_-6.7%	41.7	0.5	29.89 ± 0.07	1.424 ± 0.008
12PUF-1SiO_2_-7.9%	39.7	0.5	27.59 ± 0.01	1.491 ± 0.001

**Table 2 polymers-11-01447-t002:** Fractional free volume of PUF-SiO_2_ models (mean ± standard error).

Model	Free Volume (Å^3^)	Occupied Volume (Å^3^)	FFV (%)
37PUF-0SiO_2_ -0%	21,173.85 ± 50.05	54,427.06 ± 50.05	28.01 ± 0.07
37PUF-1SiO_2_ -2.6%	14,257.58 ± 19.32	55,946.39 ± 19.32	20.31 ± 0.03
25PUF-1SiO_2_-3.7%	7626.71 ± 35.83	38,145.95 ± 35.83	16.66 ± 0.08
18PUF-1SiO_2_ -5.3%	4422.97 ± 38.81	27,870.26 ± 38.81	13.70 ± 0.12
14PUF-1SiO_2_-6.7%	3253.46 ± 30.22	23,026.51 ± 30.22	12.38 ± 0.09
12PUF-1SiO_2_ -7.9%	2006.52 ± 14.26	18,961.32 ± 14.26	9.56 ± 0.07

**Table 3 polymers-11-01447-t003:** Micro mechanical properties of the PUF-SiO_2_ models (GPa ± standard error).

Model	37PUF-0SiO_2_-0%	37PUF-1SiO_2_-2.6%	25PUF-1SiO_2_-3.7%	18PUF-1SiO_2_-5.3%	14PUF-1SiO_2_-6.7%	12PUF-1SiO_2_-7.9%
$C_{12}$	2.7527 ± 0.3781	3.2839 ± 0.3051	3.8483 ± 0.1840	4.7356 ± 0.4571	4.9852 ± 0.5445	5.7648 ± 0.2067
$C_{11}$	7.6535 ± 0.5467	9.4918 ± 0.3322	12.1510 ± 0.2894	10.7983 ± 1.3888	12.3678 ± 1.5087	15.0725 ± 0.2480
$C_{22}$	8.5477 ± 0.2839	10.3343 ± 0.3099	10.5001 ± 0.1673	12.5241 ± 0.8194	12.8983 ± 1.4587	13.3956 ± 0.2276
$C_{33}$	9.0590 ± 0.6414	9.8329 ± 0.1831	10.6432 ± 0.1875	11.5590 ± 0.4832	11.3454 ± 1.2693	12.5315 ± 0.2640
$C_{44}$	2.5086 ± 0.1973	2.9983 ± 0.1460	3.1127 ± 0.1185	3.8134 ± 0.2170	3.8561 ± 0.2649	4.0825 ± 0.2050
$C_{55}$	2.5624 ± 0.0687	2.9176 ± 0.1447	3.1466 ± 0.0813	3.3067 ± 0.6514	3.5843 ± 1.2084	4.5313 ± 0.1048
$C_{66}$	2.1790 ± 0.0503	2.8749 ± 0.1844	2.7866 ± 0.0943	2.0627 ± 1.1979	2.8310 ± 1.1607	4.0137 ± 0.0595
E	6.28 ± 0.38	7.56 ± 0.21	7.92 ± 0.47	8.09 ± 0.33	8.94 ± 0.62	10.75 ± 0.67
K	5.20 ± 0.04	5.98 ± 0.02	7.08 ± 0.02	7.55 ± 0.04	7.64 ± 0.24	8.05 ± 0.06
G	2.42 ± 0.02	2.93 ± 0.02	3.02 ± 0.01	3.06 ± 0.07	3.42 ± 0.18	4.21 ± 0.03
γ	0.30 ± 0.01	0.29 ± 0.01	0.31 ± 0.02	0.32 ± 0.02	0.31 ± 0.01	0.28 ± 0.03
$C_{12}-C_{44}$	0.24 ± 0.18	0.29 ± 0.16	0.74 ± 0.06	0.92 ± 0.24	1.13 ± 0.28	1.68 ± 0.00

**Table 4 polymers-11-01447-t004:** Energies of the PUF–SiO_2_ interfacial interaction (kcal/mol ± standard error).

System	$E_{total}$	$E_{SiO_{2}}$	$E_{PUF}$	$E_{\mathrm{int}ercation}$	$E_{binding}$	$E_{hydrogen\;bond}$
PUF–SiO_2_	36,578,757.7 ± 6.9	36,579,050.3 ± 0.2	–225.7 ± 7.3	–66.8 ± 0.2	66.8 ± 0.2	–45.1 ± 0.8

**Table 5 polymers-11-01447-t005:** The number of hydrogen bonds for PUF-SiO_2_ models (mean ± standard error).

Model	$N_{total}$	$N_{PUF}$	$N_{nano-SiO_{2}}$	$N_{\mathrm{int}erface}$	$N_{\mathrm{int}erface}/N_{PUF}\;(\%)$
37PUF-0SiO_2_-0%	4369 ± 18	4369 ± 18	0 ± 0	0 ± 0	0.00 ± 0.00
37PUF-1SiO_2_-2.6%	4557 ± 10	4369 ± 18	27 ± 0	161 ± 8	3.69 ± 0.17
25PUF-1SiO_2_-3.7%	3046 ± 9	2880 ± 11	27 ± 0	139 ± 2	4.83 ± 0.05
18PUF-1SiO_2_-5.3%	2272 ± 4	2120 ± 6	27 ± 0	125 ± 2	5.90 ± 0.08
14PUF-1SiO_2_-6.7%	1710 ± 8	1568 ± 10	27 ± 0	113 ± 2	7.20 ± 0.08
12PUF-1SiO_2_-7.9%	1593 ± 9	1436 ± 12	27 ± 0	126 ± 3	8.75 ± 0.14

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
