# Peer review of "The Mechanical Properties of Poly (Urea-Formaldehyde) Incorporated with Nano-SiO2 by Molecular Dynamics Simulation"

_polymers, 2019, doi:10.3390/polym11091447_

Round 1
Reviewer 1 Report
I have received this manuscript for review before, and the authors tried to address the given comments of the reviewers. However, I found out some mistakes in the updated version. First, the reference section is not complete, the authors have referred to 44 references but the references section only has 36 references?!
I still have problems with the terminologies, "section 2.2" and Figure 2; this is not at all realistic. The nanoparticles could not be 0.5 nm. This scale is close to crystallographic parameters and quantum dots at the most not nanoparticles. The nanoparticles could have high specific surface area due to the presence of mesopores. This is a big fallacy to only relate the high surface area to the size. Figure 4 is more realistic and the hydrogen atoms are not present in the bulk structure, I highly recommend to implement the crystallographic constants (i.e. lines 112-113 into Figure 2 or Figure 4).
Please check other terminologies that match literature!
Author Response
Dear reviewer:
The point-by-point response has been uploaded. The manuscript has been carefully revised considering your comments. The revised parts are marked in red. We hope it can meet your expectations and requirements.
Prof. Youyuan Wang

Reviewer 2 Report
I am happy that the authors carefully re-considered their simulations and now present a much improved discussion. I now recommend their paper for publication. Just a minor thing to point out: in figures 6 and 8 it would be in order to represent equal changes in concentration by equal lengths on the x-axis.
Author Response

(The authors gave the same response as above.)

Reviewer 3 Report
This manuscript reports on molecular dynamics simulations of models of poly(urea-formaldehyde) incorporated with nano-SiO2, with the aim to evaluate the effect of the addition of this nano-filler on structural and mechanical properties of the material. Routine methods of trajectory analysis show an expected trend of the calculated observables at increasing filler concentration.
The novelty of the work is scarce and, because of several issues, I do not recommend the publication on Polymers.
The Authors use a standard simulation protocol. However, the sampling time of 40 ps is really too short to obtain a reliable evaluation of properties of polymer materials. To use a production run of tens nanoseconds allows to sample stable values, and to estimate their range of validity. In this respect, the values reported in all Tables miss errors and probably have too much meaningful digits.
The simulation approach and analysis used to determine the mechanical properties is not clear. Typically, non-equilibrium simulations allow to access this information.
The characterization of the interfacial properties with a model containing only one chain on the surface, needs that at least three independent simulations are carried out, for example using different initial 2-D conformations of the chain.
To estimate the interfacial interaction energy, were simulations of the separated components (polymer chain and nano-SiO2) carried out? In which conditions? Moreover, it is not clear how the hydrogen bond energy was calculated.
The Authors present experimental results, but the corresponding description of materials and methods is totally missing. Are these results already reported in the literature?
Page 10 line 318: is the degree of polymerization =3 ?
The text is often redundant, in particular in Section 3.4.3, where many number values with few relevance are reported.
Author Response

(The authors gave the same response as above.)

Round 2
Reviewer 1 Report
The authors revised the manuscript according to the given comments.
Author Response
Dear reviewer:
The manuscript has been carefully revised considering your comments. The revised parts are marked in red. We hope it can meet your expectations and requirements.
Prof. Youyuan Wang

Reviewer 3 Report
The revision significantly improved the manuscript, but the following points should be still addressed.
The values of parameters obtained from simulations have to be given with a range of confidence. In particular, errors have to be added to all values reported in the Tables, by explaining how such errors are estimated. Moreover, error bars have to be added in Fig. 8 and to calculated data of Fig. 9.
In Section 4.2 the number of several kinds of hydrogen bond (HB) is reported. Are these HB numbers obtained by integration of the corresponding radial distribution functions? Or are they detected directly from the trajectory by using geometric criteria for the definition of the HB occurrence (namely specific X-Y distance and XH...Y angle) ? This information should be given in the manuscript.
Being eq. (2) the experimental polymerization reaction, in such equation probably "20" should be replaced with "n", namely with a generic degree of polymerization.
For improving clarity, the sentence in page 7 lines 218-219 could be rephrased as:
"20 g of pre-polymer solution was prepared by 2.5 g of urea. On the basis of eq (2) and assuming the polymerization reaction as complete, the mass of PUF contained in the pre-polymer solution can be estimated and then the nano-SiO2 content of microparticles."
Residual English errors should be corrected. For example: Page 6 line 216: was prepared -> were prepared
Author Response

(The authors gave the same response as above.)

Round 3
Reviewer 3 Report
A computer simulation can be considered as a virtual experiment and the statistical significance of results should always be estimated. Such as in the experiment, the numerical value of an observable without its error is meaningless.
The blocking method is a typical procedure to estimate the errors of properties calculated by molecular dynamics simulations in equilibrium conditions, such as density and fractional free volume.
I recommend to use the blocking method to calculate the errors of densities reported in Table 1 (considering a suitable final interval of trajectory of the NPT-MD simulation (P= 1 bar, T=298 K, > 1500 ps)) and the errors of data reported in Table 2 and in Figure 8 (considering the trajectory of NVT-MD simulation (T=298 K, 500 ps)).
To estimate the errors for mechanical properties needs that at least two equivalent but independent models are simulated (see, for example, the elasticity modulus calculation in Soft Matter, 2016, 12, 3972-3981).
An effort in this direction should be done, so significantly increasing the meaning of this work.
Author Response
Dear reviewer:
Thank you very much for your suggestion of using blocking method to estimate the errors.
We calculated the errors of densities reported in Table 1 considering the suitable final interval of trajectory of the NPT-MD simulation (P= 1 bar, T=298 K, > 1500 ps).
We calculated the errors of FFVs reported in Table 2 and in Figure 8 considering the suitable final interval of trajectory of the NVT-MD simulation (T=298 K, 500 ps).
And we calculated the errors of hydrogen bonds numbers reported in Table 5 considering the suitable final interval of trajectory of the NVT-MD simulation (T=298 K, 500 ps).
The errors for mechanical properties were calculated based on the elasticity modulus calculation method in Soft Matter, 2016, 12, 3972-3981. And the article has been cited in our manuscript. The errors of C11, C22, C33, C44, C55, and C66 can be obtained from the result files. The errors in mechanical properties calculations were calculated as mean-square deviations from the average value, obtained by averaging over all samples and three directions x, y, z.
All the errors have been added in the manuscript.
So the result has been revised again, and the conclusion has also been improved.
Prof. Youyuan Wang

This manuscript is a resubmission of an earlier submission. The following is a list of the peer review reports and author responses from that submission.
Round 1
Reviewer 1 Report
The paper by Zhang et al. describes the results of molecular-dynamics simulations on various models of SiO2-doped PUF polymer, investigating the change in material properties such as density and mechanical moduli upon change of SiO2 concentration. The results on mechanical moduli are backed up by experimental observations. The authors report that up to around 10% SiO2 doping, the mechanical strength of the material is increased, while further doping decreases the strength; these changes are mirrored by corresponding changes in the density and the free volume of the system. To understand the mechanical properties of such nano-composites based on microscopic simulations is an timely topic, and the discussion of doped polymers fits into the scope of Polymers as a journal. I have, however, some reservations regarding the methodology used in the present paper. I can therefore recommend publication only after the points below have been sufficiently well clarified. As a general remark, the paper is well structured and in principle readable, but there are a lot of sentences where the English should be improved. Perhaps a native speaker/lector can have a look. My main concern are possible finite-size effects and a lack of sufficient statistical sampling in the simulations. For example, figure 5 suggests that the "higher SiO2" concentration, which in principle should be called "lower PUF concentration" as the authors simply decrease the number of polymeric chains around the single SiO2 nanoparticle, is obtained by reducing strongly the box size. I suspect that periodic boundary conditions have been used. But then, the assumption that the single SiO2 nanoparticle does not strongly interact with its mirror images, is not guaranteed to be true. Also, in figure 5 all simulation boxes appear to be less elongated along the third dimension, while from Table 1, they appear to be cubic with equal lengths in all Cartesian directions. Since it is not mentioned otherwise, I assume that essentially a single MD run per SiO2 concentration was performed? This appears to be insufficient to obtain reliable averaged quantities. The discussion of the radial distribution functions around figure 10 strongly suggests this: there are a lot of fluctuations still to be seen, and I suspect that at least a certain number of "peaks" that the authors discuss in conjunction with this figure could be the artifact of insufficient statistical averaging. I thus do not think that the discussion on pages 10/11 is relevant at all, unless the authors demonstrate that the peaks they are discussing there are "real". Note also that g(r) should approach unity at large distances; this approach is not seen in figure 10; one thus wonders again about possible finite-size effects. In summary, the authors need to ensure that their simulations are not influenced qualitatively by finite-size effects, before the paper can proceed to publication. This is important, because the non-monotonic trend for the mechanical properties and density that is reported, could at least in part be caused by a reduction of simulation box size. The authors also promise to give a physical explanation for the non-monotonic change in properties around 10% doping concentration, but I missed it when reading the paper. They should more clearly elucidate the point. More detailed comments follow: The interaction models for the atomic species should be given with proper references. So far, only the software (l.98) and the name of the force field (l.128) are mentioned. The authors need to tell the reader where relevant parameter values can be found in the literature. How were the force fields determined? In order to connect the simulations of the doped PUF model and the interface simulations, the authors should consider including g(r) also calculated from the former. l.115: "y is the nanoparticle number", there is no "y" in the formula. section 2.4: here, the number of MD runs averaged over needs to be given. And/or the number of configurations used to average g(r). Are the authors sure that 40000 time steps are sufficient to generate enough statistically well-separated configurations? section 3.2: is the fractional free volume different from the number density? If so, can the authors convert their mass-density values into number-density values and compare with the FFV? In fact, I found the introductino of the "fractional" free volume confusing: of course one normalizes all volumes to the total volume! l.271: the pair correlation function and the radial distribution function are, at least in some literature, different objects. (One is g(r), the other is essentially r^2 g(r).) l.305: as only one example of why I do not believe any of this discussion: a peak at r=2.75A of size 0.12107 is hardly discernible as a shoulder in the figure. Are the authors sure that it is significant? In the Author Contributions: who did the experiments? They are being discussed as being done by the present author group. If not, proper reference should be given.
Author Response

(The authors gave the same response as above.)

Reviewer 2 Report
This manuscript presents the prediction of mechanical properties of poly (urea-2 formaldehyde) doped with silica species. The topic seems interesting, however, the transition from the simulation to the experimental part is weak. The correlation should be discussed carefully. There are some wrong terminologies, which are needed to be fixed. I have some comments in detail:
1. The abstract is not well written. The aim and significance of the study are not clear. The authors also used “improve” and “weaken” in density and free volume properties, which don’t make any sense. Please rewrite the abstract more clearly!!
2. In some parts and mainly in the introduction, the authors didn’t mention properly the importance of the polymer microstructure and the interfaces. Why the MD simulation is needed? How is possible to obtain better material with the aid of simulation? How the atomic termination of the nanoparticles in the nanocomposites is important? These shortcomings should be fixed with these references:
Macromolecules 2019, 52, 807-818.
Polymer 2019, 161, 139-150
Advanced Materials 2018, 30 (4), 1703624
3. The importance of interfaces for adsorption of different species in harsh environments is important to state in the early section of the introduction, and simulation of the interfaces might also be helpful for the adsorption purposes. These references are important to be used:
Progress in Polymer Science 1996, 21 (2), 299-333
Polymer Degradation and Stability 2017, 138, 27-39
Polymer degradation and stability 2017, 136, 10-19
4. Figure 2; The simulated box for silica nanoparticles is too-small and it is in the order of crystal-lattice. This is not anymore nanoparticle, not even quantum-dots. The species in this size in experiments are showing different results. The authors also used the aggregation of this simulated box in Figure 4, which cannot again be considered as particles since the lattice is not repeating in the bulk-state like the real silica particles. The hydrogen atoms are present in the bulk of the aggregated model, and it is unlikely to have hydron in the bulk of silica nanoparticle. Therefore, the terminology and correlation to the experimental part should be reconsidered. I suggest the author use atomic termination at the interface not using the term of the nanoparticle. Please see this article:
J. Appl. Phys. 2019, 125, 045109.
5. Using the term “lattice parameters” in Table 1 is misleading, please remove it!
6. Figure 8; How the experimental results are obtained? I am wondering that the SEM image is not matching to the simulation discussion (please see previous comment)!
Author Response

(The authors gave the same response as above.)

Round 2
Reviewer 1 Report
I thank the authors for their reply, but I fear that this reply is unsatisfactory in many regards. Mainly, I was pointing out possible finite-size effects that become more dramatic as the number of polymers is reduced to mimick an increasing nanoparticle concentration. Essentially, I am concerned that in a simulation of a box of length 20A, of one particle of diameter 10A and interacting with only 5 polymeric chains, periodic boundary conditions cause a single polymeric chain to interact with effectively two nanoparticles always (the nanoparticle and one of its periodic images), which is presumably not the effect the authors are looking for. My fear may be exaggerated, but since the authors report a non-monotonic change in the structure, they should certainly check for possible finite-size effects. The authors have replied by some statements that p.b.c. "prevent any finite size effect", which is simply not true. The discussion of g(r) still contains some discussion of simulation noise (l.367), and the approach to unity at large r still is not seen. (Its lack again points out that the simulation box probably is too small.) I also do not understand how the g(r) could change so much from one version to the next of this manuscript, without the authors acknowledging this or even changing their interpretation much. It casts doubt on the analysis. I do not recommend this paper for publication.
Reviewer 2 Report
The authors addressed fairly the given comments.